# From Visual Grading and Dynamic Modulus of European Beech (*Fagus sylvatica*) Logs to Tensile Strength of Boards

**Mitja Plos** [†] , **Barbara Fortuna** [†], **Tamara Šuligoj** [†] and **Goran Turk** [*,†]

Faculty for Civil and Geodetic Engineering, University of Ljubljana, Jamova Cesta 2, 1000 Ljubljana, Slovenia; mitja.plos@fgg.uni-lj.si (M.P.); barbara.fortuna@fgg.uni-lj.si (B.F.); tamara.suligoj@fgg.uni-lj.si (T.Š.)

**\*** Correspondence: goran.turk@fgg.uni-lj.si; Tel.: +386-1-4768-614

**†** These authors contributed equally to this work.

**Abstract:** The aim of the present paper is to assess the non-destructive indicating properties of Slovenian beech (*Fagus sylvatica*) logs and correlate them with the mechanical properties of the final product, which is boards. Beech logs were visually graded according to the standard procedure and vibrational frequencies were measured. Logs were further on sawn into boards which were also non-destructively tested in wet and dry conditions. Finally, the boards were experimentally tested in tension. Special focus was directed towards visual parameters of the beech logs and their influence on the overall quality of the output material. The longitudinal natural frequencies of the logs were studied as potential indicating properties. The results showed that a majority of the visual log grading parameters do not result in good quality timber in terms of strength and stiffness properties, and only few are decisive for the final classification. The coefficient of determination of the static MOE vs. dynamic MOE of logs was $r^2 = 0.13$, whereas vs. the MOE of wet boards was $r^2 = 0.49$. Using a few visual characteristics in combination with dynamic measurements of logs and of wet boards could help to increase the yield of high quality beech wood.

**Keywords:** visual grading; machine grading; beech; tension strength; longitudinal vibration; dynamic modulus; hardwood; pregrading; moisture equilibrium; modulus of elasticity

## 1. Introduction

In the last few decades, the threat of climate change has been more and more visible everywhere around the world [1]. Among other things, this has also led to changes in the forests in Europe, where in recent years the number of Norway spruce (*Picea abies*) trees has declined [2]. According to Nothdurft, "marginal risk of tree death for 100-year-old Norway spruce trees will be doubled before the year 2100" [3]. Spruce is mainly being replaced with European beech (*Fagus sylvatica*) [2], since "beech trees and older beech trees in particular, are able to adapt to climate change and may better compensate for warmness and drought than other common European tree species" [3]. Wühlisch reported that in the years after the 2003 drought, "beech forests are highly productive and years of seed production are more frequent" [4].

Although beech is not most commonly used for construction purposes, due to the aforementioned reasons, the interest of its use in construction has grown. In countries of Central Europe, such as Slovenia, the dominance of deciduous trees over coniferous ones has gone up from 51.0% in the year 1999 to 55.3% in 2019 [5]. In accordance, spruce has gone from 32.5 to 30.4% and beech has gone up from 31.6 to 32.7%, overtaking the number one spot of the most common tree species in Slovenia from spruce [5]. However, the abundance of beech is not the only point of interest since its mechanical properties are much higher than those of spruce, as long as it is not exposed to outdoor conditions. However, there is a lot of expected rejection during the production process when using beech.

To evaluate the influence of pregrading on the mechanical properties of boards, the research started on logs of European beech. As some authors have shown on other tree

species, "pregrading of logs for strength grading improves the quality of the end product, structural timber" [6].

Ross et al. [7] measured Balsam fir (*Abies balsamea*), Eastern spruce (*Picea rubens*), and White spruce (*Picea glauca*) logs' speed of stress wave transmission and compared the measurements to the measurements of individual boards and the mean values of all the boards from a single log. For White spruce, the coefficient of determination between the longitudinally measured dynamic log modulus of elasticity (MOE) and transversally measured dynamic MOE was $r^2 = 0.5$ for individual boards. For Eastern spruce logs compared to the mean values of the boards from the same log, the value was $r^2 = 0.82$ [7].

For Norway spruce, Hanhijaarvi et al. [8] reported the coefficient of determination between the longitudinal log frequency and the static bending MOE and bending strength $f_m$ to be $r^2 = 0.37$ and $r^2 = 0.23$, respectively. For European red pine (*Pinus sylvestris*), these two values were $r^2 = 0.60$ and $r^2 = 0.35$, respectively.

In later research with Norway spruce, Hanhijaarvi et al. [9] reported the coefficient of determination between the longitudinal log frequency and the static bending MOE and bending strength $f_m$ to be $r^2 = 0.28$ and $r^2 = 0.16$, respectively. For European red pine, these two values were $r^2 = 0.37$ and $r^2 = 0.21$, respectively. In addition, they reported the coefficient of determination for Norway spruce between the longitudinal log frequency and the static tensile MOE and the tensile strength $f_t$ to be $r^2 = 0.26$ and $r^2 = 0.14$.

Similar research has been done by Rais et al. [10] where the coefficient of determination between the longitudinal natural frequency of European beech logs and wet condition boards was $r^2 = 0.39$. For dry boards, it was from $r^2 = 0.39$ to $r^2 = 0.44$ for different dimensions. When they compared the logs with the mean values of longitudinal natural frequencies of boards from a single log, the coefficient of determination was $r^2 = 0.69$ for dry and $r^2 = 0.72$ for wet boards.

Although most of the visual log grading is done primarily for quantitative purposes to enhance the yield, a look was taken into the relationship between visual characteristics of logs and mechanical properties of boards from these same logs. As opposed to spruce, European beech has fewer and, at the same time, larger branches and subsequently bigger knots as well as a larger slope of grain, both of which influence the mechanical properties of sawn timber [10]. In addition to spiral growth and log straightness, another key log characteristic, as Foley [11] pointed out, is the differentiation between sound and covered knots, since sound knots influence strength more. The measurements that Rais et al. [10] made of European beech in 2020 indicate that log taper negatively affects the wood quality. "The higher the taper was, the lower the dynamic MOE of boards was". Beech has also been shown to be more resilient to climate change through the reduction of the taper effect, as Rais et al. [12] pointed out: "With climate change, stem shapes are expected to become less tapered".

The aim of the work is to evaluate the correlations between both visual measurements and non-destructive testing and the mechanical properties of boards. The plan is to evaluate if visual grading designed for quality and quantity can also be used for the strength prediction of sawn timber. A look will be taken at each visual grading criterion, and their connections to tensile strength and static MOE will be evaluated. The dynamic MOEs of logs with the dynamic MOEs of wet and dry boards sawn from these logs will be compared with static MOEs and strengths from tensile tests on boards. The influence of the orientation of the boards within the logs on the tensile strength will be addressed. The effect the log taper has in terms of board tension strength will also be evaluated.

## 2. Materials and Methods

The material used in this work was European beech (Fagus sylvatica) from Slovenia. The research began at the sawmill where 548 logs were randomly selected over a period of two years during the winters of 2016 and 2017. All the logs were numbered. The log diameters were measured with a tree caliper and rounded down to a 1 cm accuracy. The diameters were measured 0.5 m from both ends (or 1.0 m from the end if it was a butt log)

and in the middle of the log with two perpendicular measurements. The ranges of the measurements were between 25 and 89 cm. The length of the logs was measured with a tape measure and rounded to 1 cm. The logs were 3.7 to 4.9 m long. Based on the diameter and length measurements, the taper of the logs was calculated using the mid and top diameter and the corresponding length between the two measurements (see Table 1).

**Table 1.** Characteristics of logs and boards.

|  |  | **Mean** | **STD** | **Min** | **Max** |
|---|---|---|---|---|---|
| Logs |  |  |  |  |  |
| Number of logs | *n* = 548 |  |  |  |  |
| Mid diameter | cm | 45.4 | 8.3 | 26.5 | 74.5 |
| Top diameter | cm | 44.3 | 8.2 | 26.5 | 74.5 |
| Bottom diameter | cm | 46.8 | 8.6 | 28 | 78.5 |
| Length | m | 4.1 | 0.2 | 3.7 | 4.9 |
| Boards |  |  |  |  |  |
| Nom. dimension 120 × 24 | *n* = 204 |  |  |  |  |
| Moisture content wet | % | 41.6 | 6.1 | 25 | 53 |
| Density wet | kg/m$^3$ | 797 | 65 | 686 | 1055 |
| Length | m | 3.9 | 0 | 3.9 | 3.9 |
| Width | mm | 118.1 | 2.9 | 112.1 | 126.0 |
| Thickness | mm | 24.2 | 0.2 | 23.3 | 25.0 |
| Nom. dimension 205 × 16 | *n* = 81 |  |  |  |  |
| Length | m | 4.1 | 0.1 | 3.8 | 4.3 |
| Width | mm | 203.3 | 4.0 | 181.7 | 206.3 |
| Thickness | mm | 16.0 | 0.0 | 15.7 | 16.0 |
| Nom. dimension 120 × 20 | *n* = 213 |  |  |  |  |
| Length | m | 4.0 | 0.2 | 3.2 | 4.4 |
| Width | mm | 118.8 | 3.7 | 97.3 | 123 |
| Thickness | mm | 20.0 | 0.2 | 16.7 | 20.0 |
| Nom. dimension 140 × 20 | *n* = 9 |  |  |  |  |
| Length | m | 3.3 | 0.2 | 2.7 | 3.4 |
| Width | mm | 142.1 | 0.2 | 142 1 | 42.7 |
| Thickness | mm | 20.2 | 0.5 | 19.1 | 20.5 |

For the first batch of boards (204 pieces), the moisture content was measured with the FME (model 111.602 with a hammer probe and insulated pins) resistance moisture meter from the company Brookhuis. The moisture meter is used for moisture contents below fibre saturation but is able to measure the moisture up to 100%. As many authors have shown, the influence of the moisture content on the dynamic MOE above fibre saturation is not as significant as below [10,13]. The wet density of wet boards was determined from the weight and dimensions of wet boards measured immediately after cutting the logs. The densities and moisture contents of boards during testing were determined from the weight and dimensions of the boards and from the densities and moisture contents of off cuts from the boards that were subjected to the protocol of the EN 13183-1 standard for determining the moisture content by oven dry method [14].

The requirement of the EN 1316-1 standard for qualitative classification of oak and beech [15] is that each log is placed in the class that is equivalent to the lowest class of seventeen criteria. All the visual characteristics of the logs were measured, and the corresponding grades (from A to D) for each of the characteristics were recorded. The measured characteristics were: dimensions (1), sound knots (2), unsound knots (3), covered knots (4), spiral grain (5), eccentric pith (6), simple sweep (7), ovality (8), fluting (9), transversing crack (10), other cracks (11), insect attack (12), rot (13), red heart (14), star red heart (15), stain (16), and T disease (17).

The measurements were conducted based on the guidelines of the German forest and timber councils [16]. The dimensions (1) were measured as described above. The diameter of sound knots (2) and the distances between them were measured with a tape measure. The same was done for unsound knots (3). Similarly, covered knots (4) were also measured with a tape measure. The number and distances between the knots and the branch scar quotient were measured. The branch scar quotient was measured as the seal length divided by the seal width. Spiral grain (5) was measured with two tape measures. A 1 m distance along the log was measured with one tape measure and the length of the curvature on the surface of the cross-section with the other. Eccentric pith (6) was calculated from the measured distance of the pith from the centre of the cross-section divided by the average of two perpendicular diameters of the cross-section. The simple sweep (7) was measured with a 4 m aluminium lath and a tape measure. The lath was placed on the part of the log with the biggest curvature, and the biggest offset of the log from the lath was measured. In cases of a butt log, the lath was placed 1 m from the butt end. The ovality (8) was measured as the percentage difference of the biggest and the smallest diameter of the log. Fluting (9) was measured visually as a concave part of the log. In cases of butt logs, fluting inside 1 m from the butt was not considered. The transverse crack (10) was measured with a tape measure. The mid diameter and the length of the crack inward on the surface from the log end were measured. Other cracks (11) were measured in the same manner. Insect attack (12) was visually checked. Rot (13) percent of diameter was measured with a tape measure. It was observed whether the rot was inside or outside the heart of the log. Similarly, red heart (14) and star red heart (15) percent of diameter was also measured with a tape measure. The presence of stain (16) was observed visually, as was T disease (17). When there were more instances of T disease, the distances between them were measured. The criteria of visual grading are given in the standard.

To obtain the dynamic MOE, each log was placed on two supports, using a portable laser vibrometer (PDV-100 from Polytec GMBH) to measure the response from a small sledgehammer impact, initiated by hand. The response was recorded and transformed with the fast Fourier transform (FFT) analysis into resonant frequencies with the STIG measurement system, from the company ILKON d.o.o., equipped with the Polytec PDV-100 portable laser vibrometer. These represent the longitudinal natural frequencies of the log$\nu_{log}$. From these frequencies, the dynamic $MOE_{dyn,log}$ were obtained. This was done using the general equation

$$MOE_{dyn} = \rho(2l\nu)^2,\qquad(1)$$

where $MOE_{dyn}$ is the dynamic MOE, $\rho$ is the density, $l$ is the length of the specimen, and $\nu$ is the first eigenfrequency of the specimen. The same equation was also used to determine the dynamic MOEs for both wet and dry boards.

The logs were sawn into boards of different sizes at the sawmill GG Novo mesto d.d. (SE Slovenia) using a band saw. During the sawing process, all the log numbers were transferred to the boards. This enabled us to know which boards came from which log. The number and dimensions of the boards are displayed in Table 1. From some logs, no boards were obtained for the testing that is presented in this paper. Additionally, some boards were lost or broken, or there was a problem with the measurements during testing. Due to all these reasons, there are more logs than boards in Table 1. For some of the boards, the dynamic MOE was recorded before drying and was denoted as $MOE_{dyn,wet}$. Afterwards, the boards were first naturally dried outside under a roof for 3 to 5 months and then dried in a drying kiln for approximately 1 month. The different boards from each log were used for different purposes. Some were used for bending tests, and some were used for tension tests, which will be our main comparison for the log measurements in this work. The dynamic modulus $MOE_{dyn,test}$ was recorded for all the boards after the process of drying.

To investigate the influence of the orientation and location of the board within the log on the tensile mechanical properties, for the first batch of 208 boards, the orientation as radial (R), radial-tangential (RT), or tangential (T) was recorded.

Tension tests were done in accordance with the EN 408 [17]. The specimens were loaded in tension with a clamping system with 1 m long serrated clamps. The load rate was controlled by displacement. A load cell measured the force in the dynamic piston. The displacements were measured with an LVDT (linear variable differential transformer) system placed diagonally on both sides of the boards in the middle between the clamps. The WA 20 mm LVDTs from the company HBM were used as is shown in Figure 1. The length of the LVDT measurements was five times the width of the boards. To obtain a more accurate measurement, three load cycles were performed to 40% of the expected maximal force, and the displacement on each cycle was measured. This approach proved to be correct since most of the time, the first cycle showed different stiffness than the later two, which were very similar. The third cycle was used for the calculation of the static MOE.

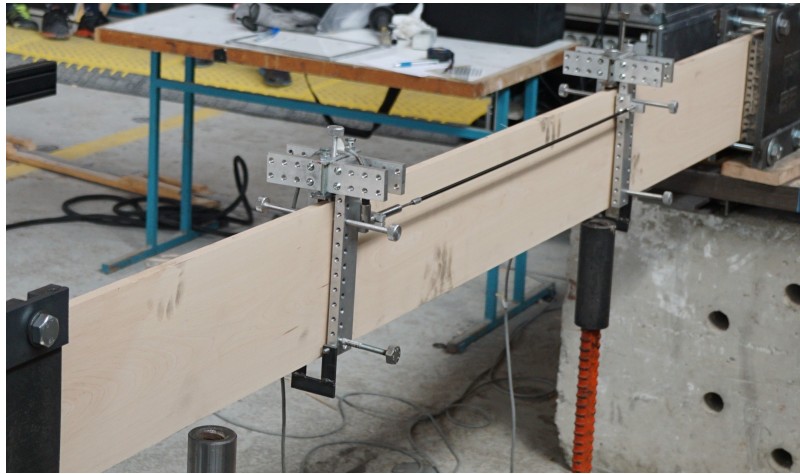

**Figure 1.** LVDT setup for static MOE measurement during tensile testing.

While conditioning the boards prior to tension tests, the moisture was lower than when dealing with a more commonly used spruce timber. Thus, specimens were taken to perform a simple moisture equilibrium test, where the specimens were left in a climate controlled chamber with a relative temperature of 20 °C and relative air humidity of 65%. Moisture equilibrium of beech was measured on 49 pieces of beech with a cross section of 120 × 20 mm and a length of 100 mm.

## 3. Results

The results of the research are given in the following sections. First, the results of visual grading of logs according to EN 1316-1 [15] are shown. Secondly, the results are presented of non-destructive dynamic MOE measurements of logs and wet and dry boards. Thirdly, the results of moisture measurements are shown. The mechanical property measurements were performed on boards that were tested in tension. The influence of the sawing pattern was also observed. At the end, the influence of the visual characteristics of logs and log taper on board strength is shown. A summary of the results is presented in Table 2.

**Table 2.** Results of logs and boards.

|  |  | Mean | STD | Min | Max |
|---|---|---:|---:|---:|---:|
| **Logs** | | | | | |
| Number of logs | $n = 548$ | | | | |
| Taper | mm/m | 3.5 | 4.9 | −21.3 | 35.1 |
| Eigenfrequency | s$^{-1}$ | 440 | 27 | 390 | 524 |
| $MOE_{dyn,log,\rho=950}$ | MPa | 12,485 | 1402 | 9065 | 16,528 |
| **Boards** | | | | | |
| Nom. dimension $120 \times 24$ | $n = 204$ | | | | |
| Eigenfrequency wet | s$^{-1}$ | 495 | 37 | 370 | 580 |
| $MOE_{dyn,wet}$ | MPa | 13,199 | 1827 | 8203 | 17,249 |
| Moisture content test | % | 7.7 | 0.6 | 6.4 | 9.0 |
| Eigenfrequency test | s$^{-1}$ | 640 | 37 | 505 | 710 |
| $MOE_{dyn,test}$ | MPa | 17,696 | 2351 | 11,403 | 22,756 |
| Static $MOE$ | MPa | 15,871 | 2910 | 5701 | 22,133 |
| Tensile strength | MPa | 66.8 | 28.5 | 6.5 | 141.5 |
| Density | kg/m$^3$ | 696 | 41 | 618 | 850 |
| Nom. dimension $205 \times 16$ | $n = 81$ | | | | |
| Moisture content test | % | 8.7 | 0.6 | 6.8 | 10.9 |
| Eigenfrequency test | s$^{-1}$ | 607 | 41 | 507 | 704 |
| $MOE_{dyn,test}$ | MPa | 17,594 | 2615 | 11,732 | 22,458 |
| Static $MOE$ | MPa | 16,080 | 3175 | 5099 | 24,607 |
| Tensile strength | MPa | 64.1 | 30.8 | 5.8 | 147.8 |
| Density | kg/m$^3$ | 724 | 62 | 549 | 878 |
| Nom. dimension $120 \times 20$ | $n = 213$ | | | | |
| Moisture content test | % | 8.7 | 0.4 | 7.7 | 9.8 |
| Eigenfrequency test | s$^{-1}$ | 609 | 42 | 470 | 767 |
| $MOE_{dyn,test}$ | MPa | 17,192 | 2475 | 9884 | 25,875 |
| Static $MOE$ | MPa | 16,720 | 3463 | 9108 | 28,799 |
| Tensile strength | MPa | 81.4 | 32.8 | 12.2 | 164.1 |
| Density | kg/m$^3$ | 724 | 47 | 620 | 898 |
| Nom. dimension $140 \times 20$ | $n = 9$ | | | | |
| Moisture content test | % | 10.8 | 0.4 | 10.4 | 11.4 |
| Eigenfrequency test | s$^{-1}$ | 728 | 46 | 660 | 820 |
| $MOE_{dyn,test}$ | MPa | 16,472 | 2039 | 13,197 | 19,440 |
| Static $MOE$ | MPa | 18,578 | 4402 | 14,445 | 25,449 |
| Tensile strength | MPa | 84.8 | 35.4 | 28.3 | 117.5 |
| Density | kg/m$^3$ | 732 | 41 | 665 | 773 |

*3.1. Visual Characteristics Measurements of Logs*

In Figure 2, it can be seen that the most influential characteristics that downgrade the logs to the lowest two grades (C and D), in order of influence, are covered knots (4), red heart (14), and sound knots (2).

As there is a debate if red heart (14) is really important since it has not been shown to have a strong influence on strength [18,19] and is therefore more or less a visual preference, two overall gradings were considered, with (Figure 3) and without (Figure 4) the inclusion of the red heart criteria. As can be seen, the only real difference is in the number of logs allocated to class C, since the standard allows red heart in both C and D classes.

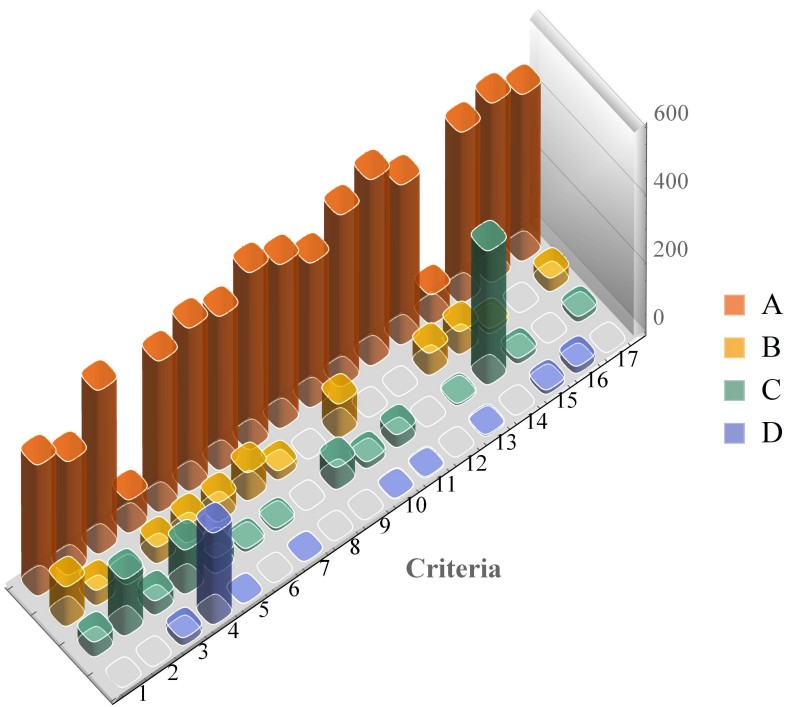

**Figure 2.** Results of visual grading by criteria (1–17).

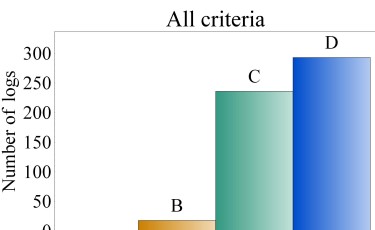

**Figure 3.** Results of visual grading with all characteristics (1–17).

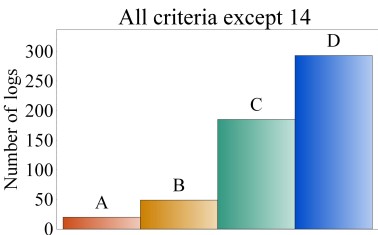

**Figure 4.** Results of visual grading with all characteristics minus the red heart characteristic (1–13, 15–17).

To see what happens when only grading based on the most influential criteria (2, 4, 9, and 14), Figure 5 was created, which is similar to the results in Figure 3, meaning that for our data, the log grading could be based on only these four criteria. The influence of knot criteria (2, 3, and 4) was also considered; therefore, the logs were graded based only on these three criteria as can be seen in Figure 6. The results are similar to those of Figure 4.

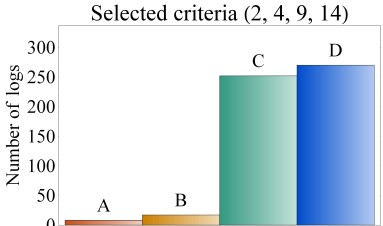

**Figure 5.** Results of visual grading with most influential characteristics (2, 4, 9, 14).

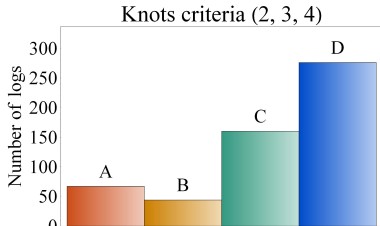

**Figure 6.** Results of visual grading with only the knot characteristics (2, 3, 4).

The cases when a criterion was the determining factor are displayed in Figure 7, together with the class the log was assigned to. From this figure, it is even clearer that the most influential criteria for this sample were 2, 3, 4, 9, and 14.

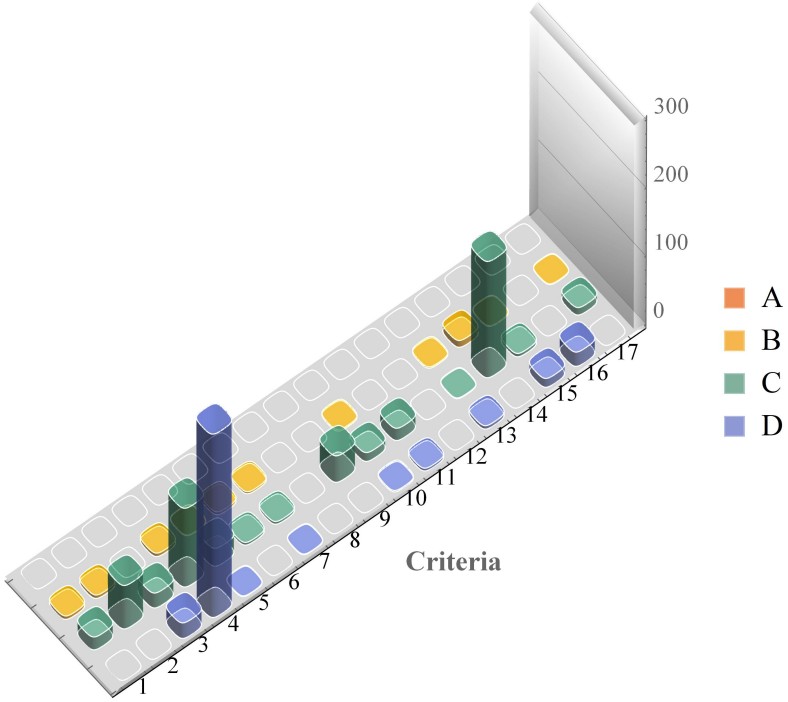

**Figure 7.** Class determining criteria.

### 3.2. Dynamic MOE Measurements on Logs and Wet and Dry Boards

All the logs were measured to acquire their longitudinal eigenfrequencies. Only the first eigenfrequency was used to determine the dynamic MOE. The results are presented in Figure 8, where the measurements for each overall visual grading class of the logs can be seen. The results of the eigenfrequency measurements do not correspond to those of visual grading, since these are two independent evaluations, one focusing on the visual characteristics mostly for the purpose of yield and the other mostly dependent on the eigenfrequency of the logs. Higher frequencies indicate higher stiffness.

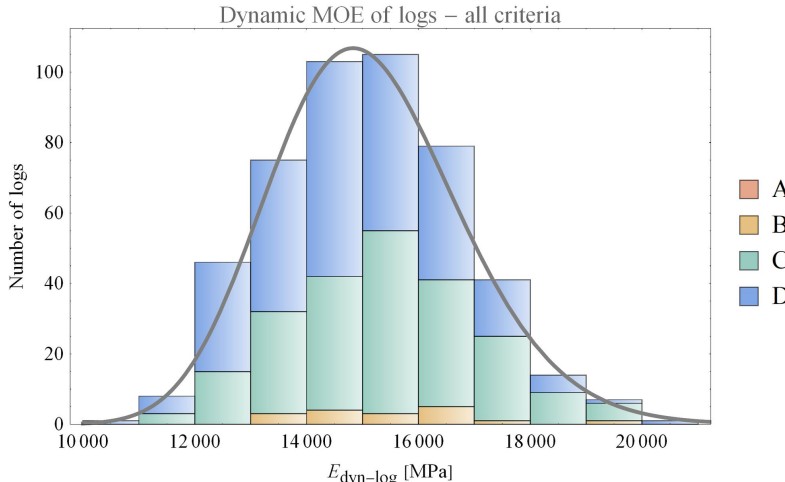

**Figure 8.** Dynamic moduli measured on logs with the PDF function (log-normal) per visual grading class.

After the logs were cut into boards, the boards were measured with the same setup. At this point, the weight and dimensions of the boards were also measured. These two measurements were later used to obtain the density. This process was repeated after the boards were first air and then kiln dried. The comparison of the dynamic MOE of boards at testing and wet boards is presented in Figure 9. The coefficient of determination was $r^2 = 0.67$. The ANOVA statistical test revealed that the linear model is statistically significant with a *p*-value lower than $10^{-6}$.

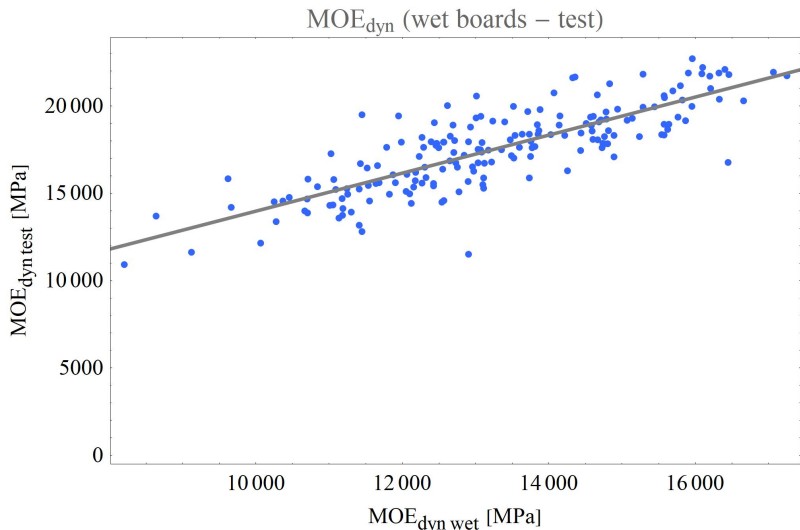

**Figure 9.** Comparison of the dynamic MOE of boards at the time of tension testing and wet boards.

For boards from both batches, the coefficient of determination between the dynamic MOEs of boards measured at the time of testing and of logs was $r^2 = 0.26$. The ANOVA statistical test revealed that the linear model is statistically significant with a *p*-value lower than $10^{-6}$.

To check the correlations between log MOE measurements and the MOEs of wet boards and boards at the time of testing, the linear regression was calculated. The results are shown in Figure 10. The coefficient of determination of wet boards and boards at testing was $r^2 = 0.28$ and $r^2 = 0.36$, respectively. The ANOVA statistical test in both cases revealed that the linear model is statistically significant with a *p*-value lower than $10^{-6}$. The results are only shown for the first batch for the comparison to wet boards, since for the second batch, the wet dynamic MOE was not measured.

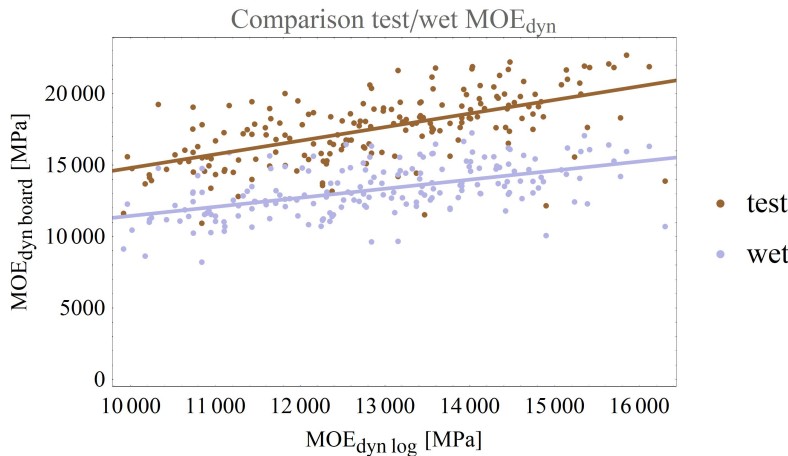

**Figure 10.** Comparison of the frequencies of logs and the frequencies of wet and dry boards.

### 3.3. *Moisture Content before and during Testing*

The results of moisture measurements on wet boards with the FME resistance moisture meter can be seen in Figure 11. The mean moisture content was 41.6%.

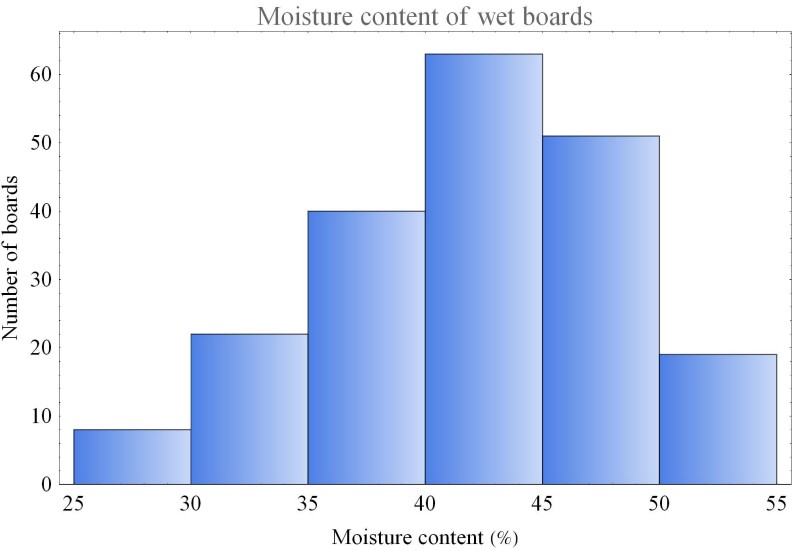

**Figure 11.** Moisture content of wet boards.

In Figure 12, the moisture content of singular boards at the time of testing can be seen. It is apparent that there is a shift in the moisture content around board 200, since this is near the end of one and start of the second batch of boards. The first batch of timber had a moisture content of 7.6%, and the second of 8.4%. Both batches together had a mean moisture content of 8.3%. The specific moisture content of each board was used to correct the results of testing to the equilibrium moisture content.

As is written in the European standards EN 338 [20] and EN 384 [21], the average moisture content "at a temperature of 20 °C and a relative humidity 65% corresponds to a moisture content of 12% for most species". Since this research is focused on beech, which is not the most common species to be used in construction, and since the mean moisture of dried boards at the time of testing for both batches was 8.3%, which is similar to the 8% ± 2% that Ehrhart et al. reported in 2018 [22], a small batch of 49 random pieces was placed in an acclimatized chamber holding standard service class 1 conditions of 20 °C and 65% relative humidity. The specimens were left in the chamber and checked every 24 h until the differences between the masses of each and every one of the specimens were less than 0.1%. The mean equilibrium moisture content of the specimens was 10.5%, with a

standard deviation of 0.21%, a minimum value of 10.0%, and a maximum value of 10.9%. The resulting moisture equilibrium values are similar to the values of 10.6% reported by Rais et al. [10].

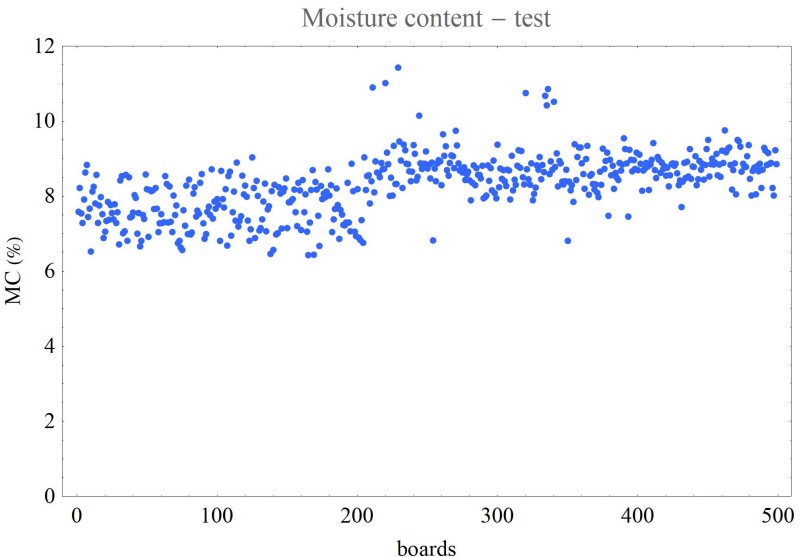

**Figure 12.** Moisture content of boards during testing.

*3.4. Tension Strength and Other Mechanical Properties of Boards*

The results of the laboratory tensile testing can be seen in Figures 13–15. The mean, STD, minimum, maximum, and characteristic values are given in Table 3.

**Table 3.** Results of grade determining properties of all boards.

|  |  | **Mean** | **STD** | **Min** | **Max** | **5th Percentile** |
|---|---|---|---|---|---|---|
| Number of boards | *n* = 482 |  |  |  |  |  |
| Static *MOE* | MPa | 16,261 | 3228 | 5099 | 28,799 | 10,868 |
| Tensile strength | MPa | 72.5 | 31.5 | 5.8 | 164.1 | 21.5 |
| Density | kg/m$^3$ | 712 | 49 | 549 | 898 | 637 |

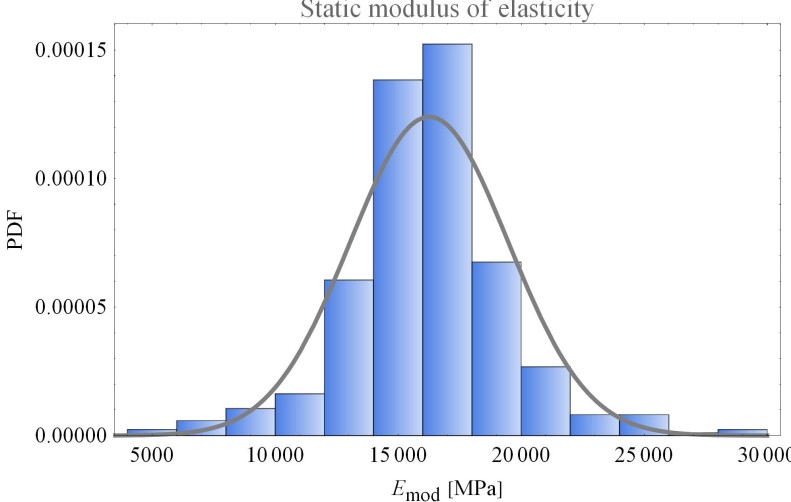

**Figure 13.** Static MOE adjusted to moisture equilibrium (*n* = 428) with normal PDF.

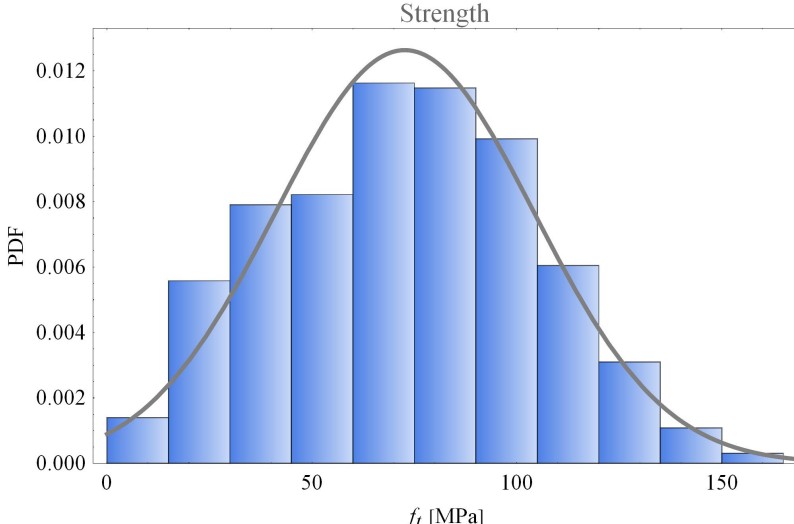

**Figure 14.** Tension strength adjusted to 150 mm width ($n = 428$) with normal PDF.

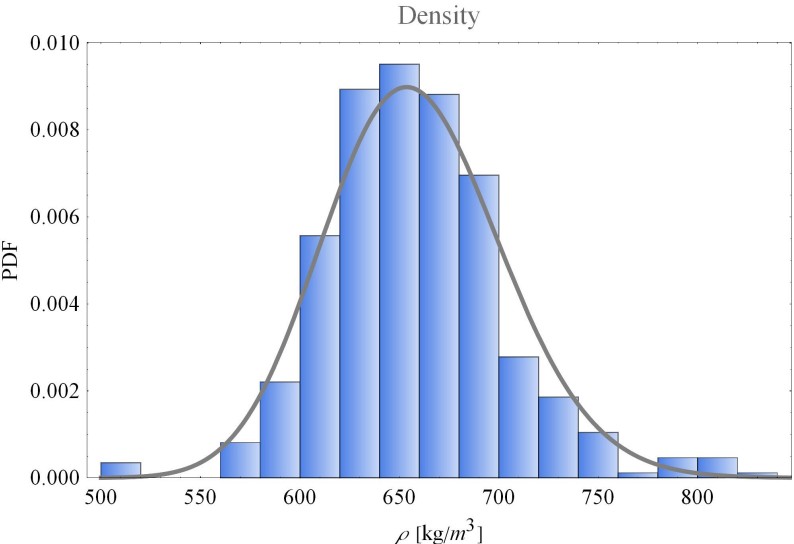

**Figure 15.** Density adjusted to moisture equilibrium ($n = 428$) with log-normal PDF.

The static MOE and the density were adjusted to the equilibrium moisture content of 10.5%. This was done in accordance with EN 384 [21] in a way that corresponds to a 1% and 0.5% change of a characteristic per 1% change in moisture content, respectively. In addition, the standard also instructs to adjust the tensile strength to a 150 mm width of the boards by dividing the strengths with the $k_h$ factor that is calculated using the following equation:

$$k_h = (150/h)^{0.2},\qquad(2)$$

where $h$, in our case, is the width of the boards.

In Figure 16, the correlation between the static MOE of dried boards and the dynamic MOE measured on logs can be seen. The coefficient of determination was $r^2 = 0.13$. The ANOVA statistical test revealed that the linear model is statistically significant with a $p$-value lower than $10^{-6}$.

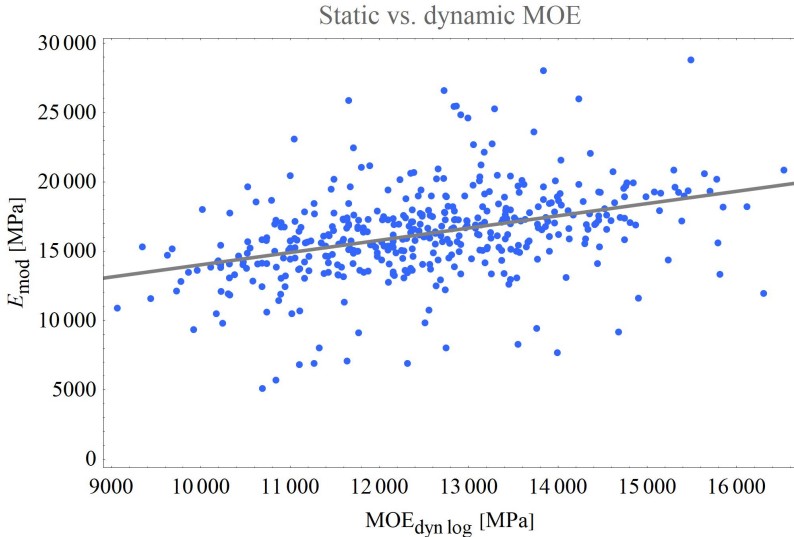

**Figure 16.** Comparison of static MOE from tension tests and dynamic MOE of logs.

In Figure 17, the comparison between the static MOE of dried boards and the dynamic MOE measured on wet boards can be seen. The coefficient of determination was $r^2 = 0.38$. The ANOVA statistical test revealed that the linear model is statistically significant with a $p$-value lower than $10^{-6}$. As can be seen in Figures 16 and 17, there were some outlier boards with fairly low static MOE values, which have a considerable negative influence on the coefficient of determination. By eliminating these eight obvious outliers, the coefficients of determination increased to $r^2 = 0.24$ and $r^2 = 0.49$, respectively. The ANOVA statistical test in both cases revealed that the linear model is statistically significant with a $p$-value lower than $10^{-6}$.

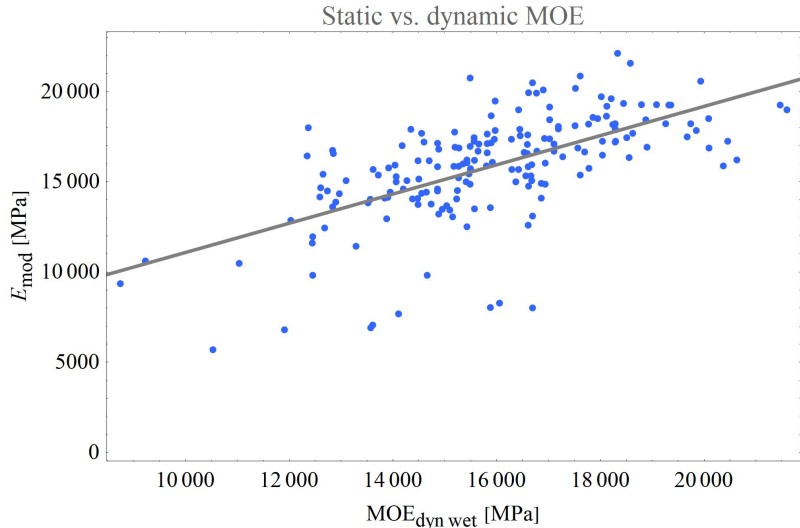

**Figure 17.** Comparison of static MOE from tension tests and dynamic MOE from boards at wet moisture content.

The dynamic MOE is assumed to have a better correlation to the static MOE than to the tensile strength, and the results confirmed this since the coefficient of determination is $r^2 = 0.05$. The results can be seen in Figure 18, where a big dispersion of the results can be seen. In this case, the $t$-test of both parameters revealed that the intercept is not significantly different from zero ($p$-value = 0.64), whereas the slope of linear regression is statistically different from zero ($p$-value lower than $10^{-6}$).

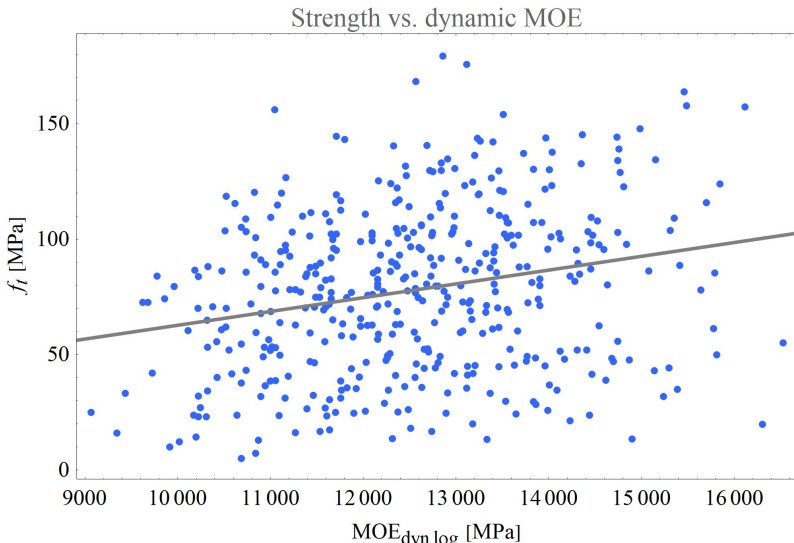

**Figure 18.** Comparison of tensile strength and dynamic MOE of logs.

As can seen in Figure 19, the coefficient of determination between the strength and the dynamic MOE of wet boards was $r^2 = 0.32$, which is much better and almost as good as with the static MOE. The ANOVA statistical test, in both cases, revealed that the linear model is statistically significant with a $p$-value lower than $10^{-6}$. This shift of $r^2$ values can partially be contributed to the smaller and more uniform sample of the first batch for which the wet boards were analysed.

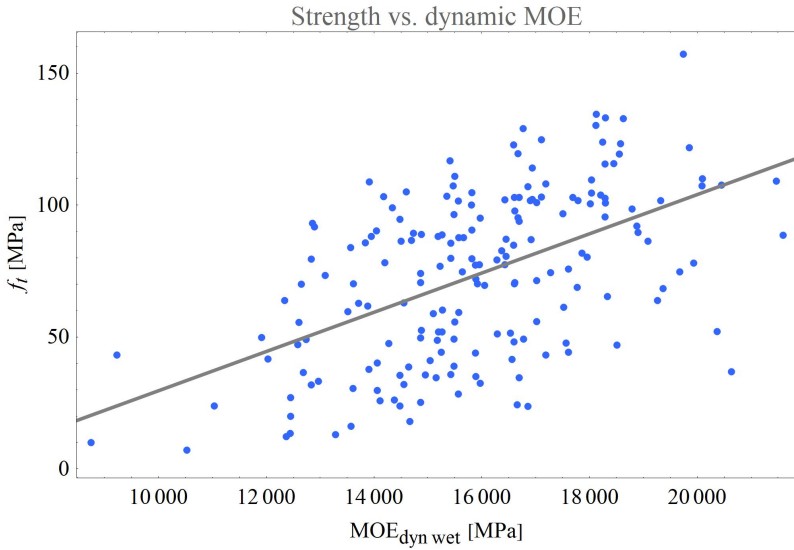

**Figure 19.** Comparison of tensile strength and dynamic MOE from boards at wet moisture content.

For 208 boards from the first batch, the orientation of the boards was measured, and based on that, the boards were allocated to three groups (R, RT, and T). The ANOVA test was performed for the tensile strength and for the static MOE. The $p$-values were 0.081 and 0.929, respectively. Since the values were both higher than 0.05, this indicates that there are no statistically significant influences of the orientation on either the tensile strength or the static MOE.

### 3.5. Influence of Visual Characteristics of Logs on Board Strength

A statistical influence analysis between the tensile strength and each class per criteria was made. The analysis was based on the ANOVA with Student–Newman–Keuls (SNK) posteriori analysis, which is a stepwise multiple comparisons procedure used to identify

sample means that are significantly different from each other. It can be concluded that for spiral grain (5), eccentric pith (6), simple sweep (7), ovality (8), fluting (9), other cracks (11), insect attack (12), rot (13), red heart (14), star red heart (15), and T disease (17), the influences were not confirmed. For example, the ANOVA test on the red heart (14) criteria had a *p*-value of 0.343, which is much more than the limit value of 0.05, indicating that the influence of red heart is not statistically significant.

Among the other characteristics, dimensions (1), sound knots (2), unsound knots (3), covered knots (4), transversing crack (10), and stain (16), there was a statistically significant difference between the classes. For example, for covered knots (4), the criteria that was shown in Section 3.1 to be most influential (see Figure 2), the *p*-value was 0.0047, confirming that the influence of this criteria is statistically significant.

The same analysis was also done for static MOE. Here, even fewer criteria were statistically different. Only dimensions (1), sound knots (2), unsound knots (3), and stain (16) had statistically significant different class values for the MOE, with *p*-values of 0.013, 0.002, 0.0003, and 0.02, respectively. When it came to the MOE, covered knots (4), the most influential criteria for strength, ceased to be statistically significant, with a *p*-value of 0.94. Only one other criterion had a higher *p*-value.

All the statistically significant criteria for both tensile strength and static MOE were consistent, with the differences being between low and high classes.

### 3.6. Taper

The results of the taper calculation were allocated to two groups, logs with a taper of 5 mm/m and less and a group with more than a 5 mm/m taper. The statistical analysis for the two groups based on static MOE showed a statistical difference with a *p*-value of 0.0065. When the same analysis was done for strength, no statistical difference was found between the the taper groups. This indicates that log taper could be a criterion worth considering to predict the stiffness of boards but not strength.

### 4. Discussion

The most influential visual characteristics for visual grading according to the EN 1316-1 [15] for data used in this study proved to be sound knots (2), covered knots (4), fluting (9), and red heart (14). Unsound knots might not seem influential, but that can be due to the fact that only a fraction of the logs had them. This effect should be considered for all non-dominant characteristics; they could be important, but due to their absence, they do not seem influential.

In terms of log pregrading to obtain boards with higher value tensile strength and static MOE, six visual characteristics had a statistically significant influence and could be used to estimate the tensile strength. In general, to produce boards with a high tensile strength and stiffness, small diameter logs with small and few knots and no stain should be used.

The coefficient of determination of the dynamic log MOE compared to the dynamic MOE of dried boards was between $r^2 = 0.26$ (both batches of boards) and $r^2 = 0.28$ (only first batch of boards). A possible reason for this relatively low value of $r^2$ may be due to the fact that boards from the same log can have different characteristics, and the comparison is made between whole log values and those of single boards. Rais et al. [10] reported values between $r^2 = 0.39$ and $r^2 = 0.44$, which is somewhat higher.

The coefficient of determination of the dynamic log MOE compared to the dynamic MOE of wet boards was $r^2 = 0.36$. In comparison, Rais et al. [10] obtained a value of $r^2 = 0.39$, which is very similar.

The moisture equilibrium of measured pieces at the temperature of 20 °C and relative air humidity of 65% was 10.5%. This indicates that equilibrium moisture content of beech is lower than that of spruce, which is usually taken as 12%.

The mean static MOE from tensile testing of beech boards was $MOE_{MEAN}$ = 16,300 MPa. The five percentile value for tensile strength was $f_t$ = 21.5 MPa, and the five percentile

value for density was $\rho = 637$ kg/m$^3$. The strength histogram was more similar to the normal than the log-normal distribution.

The coefficient of determination of the static tensile MOE compared to the dynamic log MOE was between $r^2 = 0.13$ and $r^2 = 0.24$. Hanhijärvi et al. [8,9] reported the value of $r^2 = 0.26$ for the static tensile MOE of Norway spruce, between $r^2 = 0.28$ and $r^2 = 0.37$ for the bending MOE of Norway spruce, and between $r^2 = 0.37$ and $r^2 = 0.60$ for the bending MOE of European red pine. The values seem very much dependent on species and somewhat dependent on the type of testing. Due to larger knots and greater grain deviation in beech than in Norway spruce or European red pine, the prediction of the static MOE by the dynamic MOE is less accurate. If the dynamic MOE measurements of wet boards are used instead of logs, the coefficient goes up to between $r^2 = 0.38$ and $r^2 = 0.49$.

The coefficient of determination of the tensile strength compared to the dynamic log MOE was $r^2 = 0.05$. Hanhijärvi et al. [8,9] reported the value of $r^2 = 0.14$ for the static tensile MOE of Norway spruce, between $r^2 = 0.23$ and $r^2 = 0.35$ for the bending MOE of Norway spruce, and between $r^2 = 0.16$ and $r^2 = 0.21$ for the bending MOE of European red pine. These are similar results to the ones with the static MOE, but all are slightly lower. If the dynamic MOE measurements of wet boards are used instead of logs, the coefficient goes up to $r^2 = 0.32$.

Although the sawing pattern is often an important parameter when it comes to other tree species and other types of tests, for beech boards in tension, the growth ring orientation was shown not to be statistically significant.

## 5. Conclusions

In general, pregrading could have some benefits in the production process in terms of strength grading and cost and rejection reduction. Using some of the characteristics from the visual standard in combination, dynamic measurements of logs and of wet boards could help to increase the yield of beech wood. Although the effect is not as good as with some other tree species, it is still worth considering. In the future, it might be worth investigating the relationship between visual characteristics, vibrational frequencies of logs and wet boards, and grade-determining properties with a decision tree method examining all the measured parameters, even taper. Further investigation should also be made to repeat these studies on bending tests.

The results of this study provide the basis for small sawmills that grade logs visually and cannot afford expensive state-of-the-art scanning machines. These small sawmills could visually grade logs and use an affordable dynamic MOE measurement system to measure both the logs and, later, the wet boards.

**Author Contributions:** Conceptualization, M.P. and G.T.; methodology, M.P. and G.T.; software, T.Š. and G.T.; formal analysis, G.T.; investigation, M.P., B.F. and T.Š.; resources, M.P. and G.T.; data curation, B.F., T.Š. and M.P.; writing—original draft preparation, M.P.; writing—review and editing, G.T., B.F. and T.Š.; supervision, G.T.; project administration, M.P. and B.F.; funding acquisition, M.P. and G.T. All authors have read and agreed to the published version of the manuscript.

**Funding:** This research was funded through the EU HARDWOODS project funded by the WoodWisdom-net research program under the ERA-NET Plus scheme of the Seventh Framework Program (FP7) and the Slovenian Ministry of Education, Science, and Sport, and through the TIGR4SMART project funded by the European Regional Development Fund; the Slovenian Ministry of Education, Science, and Sport; and the Faculty of Civil and Geodetic Engineering of the University of Ljubljana.

**Institutional Review Board Statement:** Not applicable.

**Informed Consent Statement:** Not applicable.

**Data Availability Statement:** The raw data presented in this study are available on request from the corresponding author.

**Acknowledgments:** The authors are thankful to Lorenz Breinig, Žiga Krofl, Urška Bajc, Urška Blumauer, Robert Pečenko, Jan Dobnikar, and Erik Raspet for their help during the testing and the measuring process. The authors are also thankful to Andrej Mikec from the company GG Novo Mesto d.d. for his help during log measurements and cutting, and Ivan Lakner from the company ESOL d.o.o. for his work during the process of drying the boards. The authors are also thankful to the company ILKON d.o.o. for their help with the dynamic MOE measurements with their STIG machine and software.

**Conflicts of Interest:** The authors declare no conflict of interest.

## Abbreviations

The following abbreviations are used in this manuscript:

| | |
|---|---|
| ANOVA | Analysis of variance |
| FFT | Fast Fourier transform |
| LVDT | Linear variable differential transformer |
| MOE | Modulus of elasticity |
| PDF | Probability density function |
| SNK | Student–Newman–Keuls method |
| STD | Standard deviation |

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
