# Peer review of "From Visual Grading and Dynamic Modulus of European Beech (Fagus sylvatica) Logs to Tensile Strength of Boards"

_forests, doi:10.3390/f13010077_

Round 1

Reviewer 1 Report

The paper is very interesting, but need to be improved. Specific comments are in the file.

Author Response

All the responses are summarized in the attached file. Please see the attachment.

Reviewer 2 Report

Manuscript is lacking several relevant considerations when it comes to properties of wood:

  • It is not clear how the tracking of boards coming from specific log was performed. In table 1 number of logs and number of boards do not match. If tracking was not made, it should be clearly presented to the reader and not misleading the reader.
  • Properties of wood highly depend on the orientation of the board as well as location within the log. Authors should add clear explanation how the boards were cut and from which part of the log the boards were from. If this influential parameter was not considered in the design of experiment phase of the research, the manuscript is not providing any meaningful contribution to science.
  • The moisture content of wood is one of the most influential parameter on the mechanical properties of wood. Statement that the moisture content is assumed to be 12% is wrong. The authors need to provide details on drying procedure and measured moisture content. Additionally it is not clear if conditioning at 20C and 65% RH was performed long enough to reach the equilibrium. Authors need to provide details, including how they assured the equilibrium was reached.
  • Discussion section is just summarizing the results previously presented and doesn't add any value. The discussion should be rewritten.
  • The criteria of visual grading should be described and not just citing the standard. It is the main part of the manuscript and reader should not search for a standard in order to understand the experiment that was performed.
  • Authors should add the discussion how the applied research is relevant for future development and utilization of beech wood compared to the state of the art scanning technologies that are utilized in advanced sawmills.

Author Response

(The authors gave the same response as above.)

Round 2

Reviewer 2 Report

Manuscript has been significantly improved.